# The methyltransferase N6AMT1 participates in the cell cycle by regulating cyclin E levels

**Margit Mutso●\*, Baiba Brūmele, Evgeniia Serova, Fred Väärtnõu, Mihkel Suija, Reet Kurg**

Institute of Technology, University of Tartu, Tartu, Estonia

\* margit.mutso@ut.ee

## Abstract

The methyltransferase N6AMT1 has been associated with the progression of different pathological conditions, such as tumours and neurological malfunctions, but the underlying mechanism is not fully understood. Analysis of N6AMT1-depleted cells revealed that N6AMT1 is involved in the cell cycle and cell proliferation. In N6AMT1-depleted cells, the cell doubling time was increased, and cell progression out of mitosis and the G0/G1 and S phases was disrupted. It was discovered that in N6AMT1-depleted cells, the transcription of cyclin E was downregulated, which indicates that N6AMT1 is involved in the regulation of cyclin E transcription. Understanding the functions and importance of N6AMT1 in cell proliferation and cell cycle regulation is essential for developing treatments and strategies to control diseases that are associated with N6AMT1.

## Introduction

Human N-6 Adenine-Specific DNA Methyltransferase 1 (N6AMT1) is a small and evolutionally conserved protein that has been shown to be active in organisms ranging from prokaryotes to humans [1–6]. When N6AMT1 forms a complex with TRMT112, it functions as a protein methyltransferase [7–10]. Recently, N6AMT1 has been associated with different pathological conditions, such as tumours [11–15], neurological malfunctions [16, 17], and diabetes [18, 19], as well as embryonic development [17] and poison metabolism [18, 20–23]. Therefore, it is critical to understand in detail how N6AMT1 is involved in different cellular functions, both in cancer and noncancerous cells.

The most well-characterized substrates of the N6AMT1:TRMT112 methyltransferase complex are histone 4 and eukaryotic Release Factor 1 (eRF1), in which a methyl group is added to the lysine 12 [7] or glutamine 185 [9] residues, respectively. The methylation of eRF1 and H4K12 is associated with the proliferation rate of tumour cells [7, 24]. eRF1 and 11 other proteins are methylated by the N6AMT1:TRMT112 complex at the same GQx3R motif [25]. Structural analysis indicates that N6AMT1 utilizes the same pocket to methylate both lysine and glutamine [9].

Initially, N6AMT1 was described as a DNA N6-methyladenin (6mA) methyltransferase. In vitro data about DNA methylation by N6AMT1 are thus far very contradictory [12, 26, 27]. Higher levels of DNA N6-methylation are related to carcinoma development and progression

tuit.ut.ee) Basic funding grant PLTTI20915, award to RK and Estonian Research Council (https://etag.ee/) grant PSG923, award to MM. The funders had no role in study design, data collection and analysis, decision to publish, or preparation of the manuscript.

**Competing interests:** The authors have declared that no competing interests exist.

in humans. N6AMT1 data from tumour cells and mouse models show a correlation between higher levels of DNA m6A and N6AMT1 DNA methyltransferase activity [13–15]; however, in some other cases, no correlation was found [28]. Since N6AMT1 is a potential DNA N6 methyltransferase, it has been suggested that N6AMT1 could be used as a tumour marker or therapeutic target in cancer. Indeed, N6AMT1 depletion clearly inhibits cell proliferation and invasion [7, 13–15, 24, 29]. RNA sequencing data from multiple research groups indicate that depletion of N6AMT1 affects the expression of genes that are involved in cell death, proliferation and cell cycle regulation [7, 11, 24, 29], which can all contribute to tumour progression.

The cell cycle is mainly regulated by the cyclin and cyclin-dependent kinase (CDK) complexes that are involved in the regulation of gene transcription and mRNA processing. CDK activity is regulated by phosphorylation, interactions with inhibitor proteins, and most importantly, via the expression of cyclins. Some of the main and most well-described cyclins are cyclin A, cyclin B1 and cyclin E, and they are expressed in a cell cycle phase-specific manner [30, 31].

Recently, it has been demonstrated that arginine methyltransferases participate in cell cycle regulation by methylating different cell cycle-related regulators, such as CDK4, a key regulator of the G1-S transition [32], the inner centromere protein [33] or transcriptional regulation factors such as E2F [34] and others [35]. Similarly, different lysine methyltransferases have been shown to affect various cell cycle-related regulators [36–40].

The aim of the current study was to investigate the association of N6AMT1 with the cell cycle. The depletion of N6AMT1 in U2OS cells led to decreased cell growth rates, delayed progression of cells through all the phases of the cell cycle, and caused high sensitivity to serum depletion. In N6AMT1-depleted cells, cyclin E, but not cyclin A or cyclin B, was downregulated at the protein and mRNA levels. The downregulation of cyclin E directly or indirectly affected the progression of cells through all the phases of the cell cycle and cell growth. To the best of our knowledge, this is the first study to demonstrate the involvement of N6AMT1 in regulating the levels of cyclin E, which is essential for normal cell cycle progression.

## Material and methods

### Cell culture

Human osteosarcoma (U2OS) cells were obtained from ATCC (American Type Culture Collection; Manassas, VA, USA), and ΔN6AMT1#1, ΔN6#1+N6AMT1 and ΔN6#1+-N6AMT1-eGFP cell lines (generated in the current study) were grown in complete culture media (Iscove's Modified Dulbecco's Media (IMDM) supplemented with 10% foetal calf serum (FCS) (Gibco, Thermo Fisher Scientific, Waltham, MA, USA), 100 U/mL penicillin and 100 μg/mL streptomycin). The cells were incubated at 37°C in a 5% $CO_2$ environment.

### Construction of N6AMT1 knockout cell lines

N6AMT1-knockout cell lines were generated by using the pSpCas9(BB)-2A-Puro (PX459) V2.0 plasmid vector (Cat no 62988, Addgene, USA). Guide RNA was designed to target exon 1 of the N6AMT1 gene. Primers (F 5' CACCGCGACGTGTACGAGCCCGCGG3, R 5' AAACCC GCGGGCTCGTACACGTCGC) were ordered from Microsynth (Balgach, Switzerland). Oligos were annealed and inserted into the pSpCas9(BB)-2A-Puro (PX459) V2.0 plasmid vector. The obtained plasmid was transfected into U2OS cells using Lipofectamine 2000 reagent following the manufacturer's protocol. Twenty-four hours after transfection, 5 μg/ml puromycin selection was initiated, and puromycin was removed after 4 days. Approximately two weeks after transfection, single cell colonies were selected, and the absence of N6AMT1 protein expression was confirmed by immunoblotting and Sanger sequencing.

## Proliferation assay

Cell proliferation was determined by xCELLigence Real-Time Cell Analyzer (RTCA SP Instrument, Version November 2009, ROCHE, USA) according to the manufacturer's protocol. Briefly, $5 \times 10^3$ U2OS, U2OS ΔN6AMT1#1–10 and U2OS ΔN6AMT1#1–24 cells were seeded in 16-well E-plates (Agilent) and placed in an xCELLigence Real-Time Cell Analyzer. Cell indices were automatically measured every 30 min for 7 days. To analyse cell proliferation upon starvation, first, the cells were seeded in 16-well E-plates in growth media, and 24 h later, the medium was changed to serum-depleted medium; this was considered the 0 h timepoint. The cell index was measured every 30 min for 120 h. Relative velocities represent the changes in the cell index over time.

## Cell doubling time

U2OS, U2OS ΔN6AMT1#1–10 and U2OS ΔN6AMT1#1–24 cells ($1 \times 10^5$) were seeded in 60 mm plates and incubated for 0, 24 h, 48 h, 72 h, or 96 h. For each timepoint, the cells were counted using Trypan Blue Stain (0.4%) and Countess™ Automated Cell Counter (T10282, Invitrogen™, Germany), and the cell doubling time was calculated.

## Cell synchronization

Cells were synchronized in different cell cycle phases. Unsynchronized cell populations were collected 18 h post seeding. The starved cell population (G0/G1 arrested cells) was first seeded in complete growth media to allow the cells to attach. After cell attachment, the medium was replaced with serum-depleted medium, and samples were collected 24 h later. Cells were arrested in S phase with double thymidine treatment. Cells were treated with 2 mM thymidine for 16 h and then released for 6 h in complete culture media, followed by another 16 h treatment with 2 mM thymidine, after which the cells were collected. The mitotic cell population was obtained after incubation with 50 ng/ml nocodazole for 10 h, followed by mitotic shaking. Only mitotic cells were collected.

## Flow cytometry

Cells were synchronized as described above. Following treatment, the cells were collected and fixed with 80% ethanol for 30 min, followed by 200 μg/ml RNAse A and 50 μg/ml propidium iodide treatment for 40 min. The cells were analysed with an Attune NxT Flow Cytometer (Thermo Fischer Scientific, Waltham, MA, USA) according to the manufacturer's instructions. Data analysis was performed with FlowJo (Beckton Dickinson, Franklin Lakes, NJ, USA) software. The experiment was repeated three times, and the average % of the cells in each cell cycle phase was calculated.

## Sample preparation for normalized western blotting assay

Cells were synchronized as described above. Following treatment, the cells were lysed in RIPA buffer (150 mM NaCl, 1,0% Triton X-100, 0,5% sodium deoxycholate, 0,1% SDS, 50 mM Tris (pH 8,0), 1 mM EDTA, 1x protease inhibitors). The total protein concentration was measured using a BCA Protein assay kit (PierceTM BCA Protein Assay Kit 23227, Thermo Scientific, USA) according to the manufacturer's instructions. A total of 2.5–10 μg of total protein was loaded per well for immunoblotting.

## Western blotting

Viable cells in PBS buffer or cell lysates in RIPA buffer were reduced with DTT in 2x Laemmli buffer and denatured at 100˚C. Western blotting was carried out as previously described [41]. Target proteins were detected using anti-α-tubulin (1:10000, T5168, Sigma–Aldrich, St. Louis, MO, USA), anti-TRMT112 (1:500, sc-398481, Santa Cruz Biotechnology, Dallas, TX, USA), anti-Cyclin B1 (1:500, sc-245, Santa Cruz Biotechnology, Dallas, TX, USA), anti-Cyclin E (1:500, sc-247, Santa Cruz Biotechnology, Dallas, TX, USA), anti-Cyclin A (1:500, sc-271682, Santa Cruz Biotechnology, Dallas, TX, USA), anti-N6AMT1 (1:1000, CQA1550, Cohesion Biosciences, London, UK), and anti-GAPDH (1:2000, sc-32233, Santa Cruz Biotechnology, Dallas, TX, USA) antibodies. Goat anti-rabbit-HRP (1:10000, 31460, Invitrogen, Carlsbad, CA, USA) and goat anti-mouse-HRP (1:10000, 31430, Invitrogen, Carlsbad, CA, USA) were used as secondary antibodies.

## RNA interference

Interfering RNA against N6AMT1(siN6AMT1), targeting exon3 (`AAUGUGAACUUUGUUAC AGCG`), exon 5 (`UUUAAUGGUAACUAAAUAGAA`) and 3'UTR (`ACUAAGUCUAAUGUAGUUCCU`) of siN6AMT1, were ordered from Microsynth (Balgach, Switzerland). As a control, the negative control siRNA used in this study was previously described [1]. U2OS cells were transfected with 250 pmol of control and N6AMT1 siRNA pool using Lipofectamine RNAiMAX (Thermo Fisher Scientific) following the manufacturer's protocol. The cells were lysed in RIPA buffer 48 h later, the total protein amount was measured, and 10 μg of total protein was analysed by Western blotting assay.

## Quantitative PCR

Total RNA was isolated from cells using the Zymo Quick-RNATM MiniPrep kit (Zymo Research, R1055) according to the manufacturer's protocol. One microgram of total RNA was used to synthesize cDNA with the RevertAid First Strand cDNA Synthesis Kit (Thermo Fisher Scientific, K1621) using random hexamer primers according to the manufacturer's protocol. Quantitative PCR (qPCR) was performed using a LightCycler 480 (Roche) with HOT FIRE-Pol® EvaGreen® qPCR Mix Plus (Solis BioDyne, 08-24-0000S) using the following conditions: 95˚C for 12 min followed by 40 cycles of 95˚C for 15 sec, 60˚C for 15 sec and 72˚C for 20 sec. Specific primers were used to measure the expression of Cyclin A2 (`F 5' CTCTACAC AGTCACGGGACAAAG, R 5' ACCACCCTGTTGCTGTAGCCAA`), Cyclin B1 (`F 5' GACCT GTGTCAGGCTTTCTCTG, R 5' GGTATTTTGGTCTGACTGCTTGC`), Cyclin E1 (`F 5' TG TGTCCTGGATGTTGACTGCC, R 5' CTCTATGTCGCACCACTGATACC`), and GAPDH (`F 5' GTCTCCTCTGACTTCAACAGCG, R 5' ACCACCCTGTTGCTGTAGCCAA`). Relative fold changes were calculated using the 2–ΔΔCt method [42] with GAPDH as the housekeeping gene.

## Cell synchronization timepoints

For cell cycle progression out of mitosis, $2 \times 10^5$ mitotic cells were seeded per 60 mm plate in complete culture media. The cells were incubated for 0 h, 1 h, 2 h, 3 h, 6 h, 9 h, 12 h, 15 h, and 18 h in complete medium and collected for analysis by Western blotting assay. At the 12 h timepoint, 50 ng/ml nocodazole treatment was added to the remaining plates and maintained until the cells were collected at the last timepoint.

For cell cycle progression out of the S phase, $1 \times 10^5$ mitotic cells were seeded per 60 mm plate. The cells were incubated in complete culture media for 6 h, followed by 2 mM thymidine

treatment for 16 h. Next, cells were washed with PBS, timepoint 0 was collected, and media was changed to complete culture media in the rest of the plates. The cells were collected at 0 h, 3 h, 6 h, 9 h, 12 h, 15 h, 18 h, 21 h, and 24 h post release. At the 15 h timepoint, 50 ng/ml nocodazole treatment was added to the remaining plates and maintained until the cells were collected at the last timepoint.

For cell cycle progression out of the G0/G1 phase, $2 \times 10^5$ cells per 60 mm plate were seeded. After 10 h, the cells were washed with PBS and incubated with serum-depleted medium. After 24 h, timepoint 0 was collected, and for the rest of the cells, the medium was changed to complete growth medium. The cells were collected at 0 h, 3 h, 6 h, 9 h, 12 h, 15 h, 18 h, 21 h and 24 h post release. At the 15 h timepoint, 50 ng/ml nocodazole treatment was added to the remaining plates and maintained until the cells were collected at the last timepoint. All samples were collected in 50 μl of 2x Laemmli buffer and analysed by Western blotting as previously described.

### Statistical analyses

Western blotting images were quantified using ImageJ software. The expression of all the proteins of interest was normalized against the expression GAPDH by calculating the protein of interest and GAPDH expression ratio. Graphs were prepared, and statistical analysis was performed using multiple unpaired t test, Holm-Šidák method with GraphPad Prism 8.4.3 software.

### Results

#### N6AMT1 depletion leads to decreased cell proliferation rates in U2OS cells

The first aim of this study was to investigate the role of N6AMT1 in the proliferation of a stable N6AMT1-knockout cell line. N6AMT1-knockout cells were established from U2OS cells using the CRISPR/Cas9 gene editing system, targeting exon 1 in the N6AMT1 gene. Two single-cell colonies, named ΔN6AMT1#1 and ΔN6AMT1#2, were selected for further characterization. N6AMT1 knockout was confirmed by sequencing, and a clear depletion of the N6AMT1 protein was observed in both colonies (Fig 1A). The proliferation rate of the N6AMT1-depleted

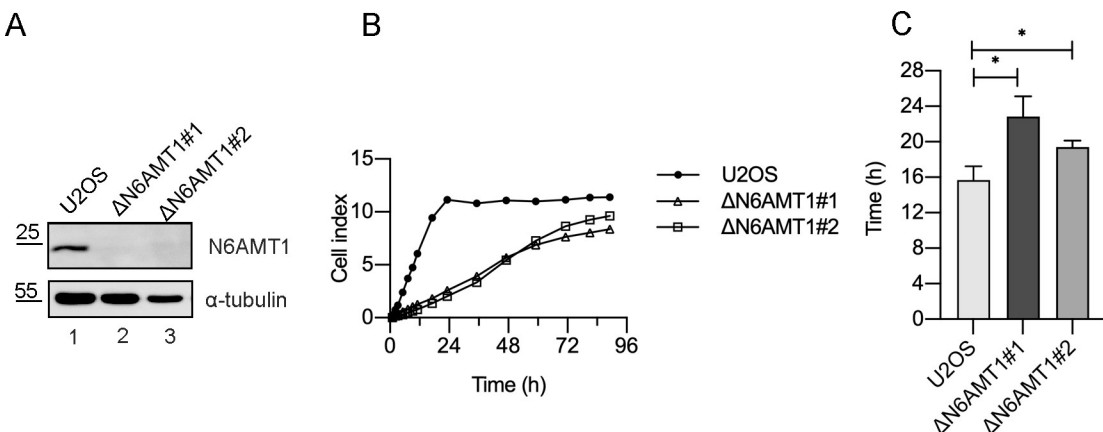

**Fig 1. N6AMT1 impacts the cell proliferation rate.** (A) The U2OS and N6AMT1-depleted cell lines (ΔN6AMT1#1 and ΔN6AMT1#2) were analysed via Western blotting with antibodies against N6AMT1 and α-tubulin. (B) The proliferation rate of the U2OS, ΔN6AMT1#1 and ΔN6AMT1#2 cell lines was determined using an xCELLigence Real-Time cell analyser, and data were recorded for a total of 7 days. (C) Doubling time of the U2OS, ΔN6AMT1#1 and ΔN6AMT1#2 cell lines. The average of three biologically independent samples is shown.

cells was analysed using the eXCelligence system, which showed a clear reduction in the growth rate compared to U2OS cells (Fig 1B). The cell growth rate reduction was also detected in partial N6AMT1 k/o HEK293 cells (S1A Fig). The population doubling time was also measured; for U2OS cells, it was approximately 15 hours, while for ΔN6AMT1#1 and ΔN6AMT1#2 cells, the doubling times were 23 hours and 19 hours, respectively (Fig 1C). These findings suggest that N6AMT1 plays a role in cell proliferation, as its depletion resulted in reduced growth rates and increased doubling time.

## Depletion of N6AMT1 disrupts the U2OS cell cycle

Both the ΔN6AMT1 #1 and #2 cell lines showed clearly reduced growth rates; therefore, it was investigated whether the absence of N6AMT1 has a significant effect on cell cycle distribution compared to U2OS cells.

There was a notable increase in the percentage of cells in the G1 phase in the unsynchronized ΔN6AMT1 #1 cell line compared to the U2OS cell line (Fig 2B). The cell population in the G1 phase increased from approximately 50% in the unsynchronized U2OS cells to approximately 65% in the ΔN6AMT1#1 cell line (Fig 2B). Additionally, there was a slight decrease in the percentage of cells in the G2/M phase, from approximately 25% in U2OS cells to approximately 15% in the ΔN6AMT1#1 cell line. Moreover, the population in the S phase was similar in both the U2OS and ΔN6AMT1#1 cell lines (Fig 2B). None of the observed effects were significantly different. To analyse the cell cycle distribution in detail, U2OS and ΔN6AMT1 #1 cells were synchronized in different cell cycle phases using serum depletion [43], thymidine treatment [44] or nocodazole treatment [45] (Fig 2A). Nocodazole-treated U2OS cells exhibited the expected G2/M phase arrest profile (Fig 2C). Approximately 70% of the U2OS cells were arrested in the G2/M phase, while in ΔN6AMT1#1 cells, the percentage of cells in the G2/M phase was only approximately 50%. The most notable difference was observed in the cell population in the G1 phase; approximately 20% of ΔN6AMT1#1 cells were found to be in the G1 phase and only 7% of U2OS cells remained in the G1 phase. No difference was observed between the U2OS and ΔN6AMT1#1 cell lines after thymidine treatment (Fig 2D) or serum depletion (Fig 2E). These findings suggest that the absence of N6AMT1 may affect the progression of cells through the G1 phase of the cell cycle.

## N6AMT1 depletion downregulates cyclin E but not cyclin A and B

The cell cycle is regulated by different cyclins; therefore it was analysed whether the absence of N6AMT1 affects cyclin levels and therefore contributes to cell cycle progression.

Cyclin A, B and E levels were measured in the cells in different phases of the cell cycle: unsynchronized cells, cells synchronized in the G0/G1 and S phases and mitotic cells that were obtained by mitotic shake. The same amount of total protein was loaded on a Western blotting gel, and the cyclin protein levels were compared between U2OS and ΔN6AMT1#1 cell lines.

The most prominent difference between the U2OS and ΔN6AMT1# cell lines was observed in the cyclin E level. In unsynchronized cells and in the serum-starved population (Fig 3A, Lanes 1–4), the cyclin E levels were reduced by approximately 2-fold in the ΔN6AMT1#1 cell line (Fig 3B). The difference was even more significant in thymidine-treated cells (Fig 3A, Lanes 5–6, Fig 3B). In the mitotic population, the cyclin E level remained undetectable in both cell lines, as expected (Fig 3A, Lanes 7–8). There was no difference in the levels of cyclin A and B in unsynchronized, S phase-arrested or mitotic cells (Fig 3A, Lanes 1–2,5–8, Fig 3C and 3D). In the serum-depleted cells, the cyclin A and B levels were significantly higher in the ΔN6AMT1#1 cell line than in the U2OS cells (Fig 3A, Lanes 3–4, Fig 3C and 3D). No difference was observed in TRMT112, a cofactor that is essential for N6AMT1 protein

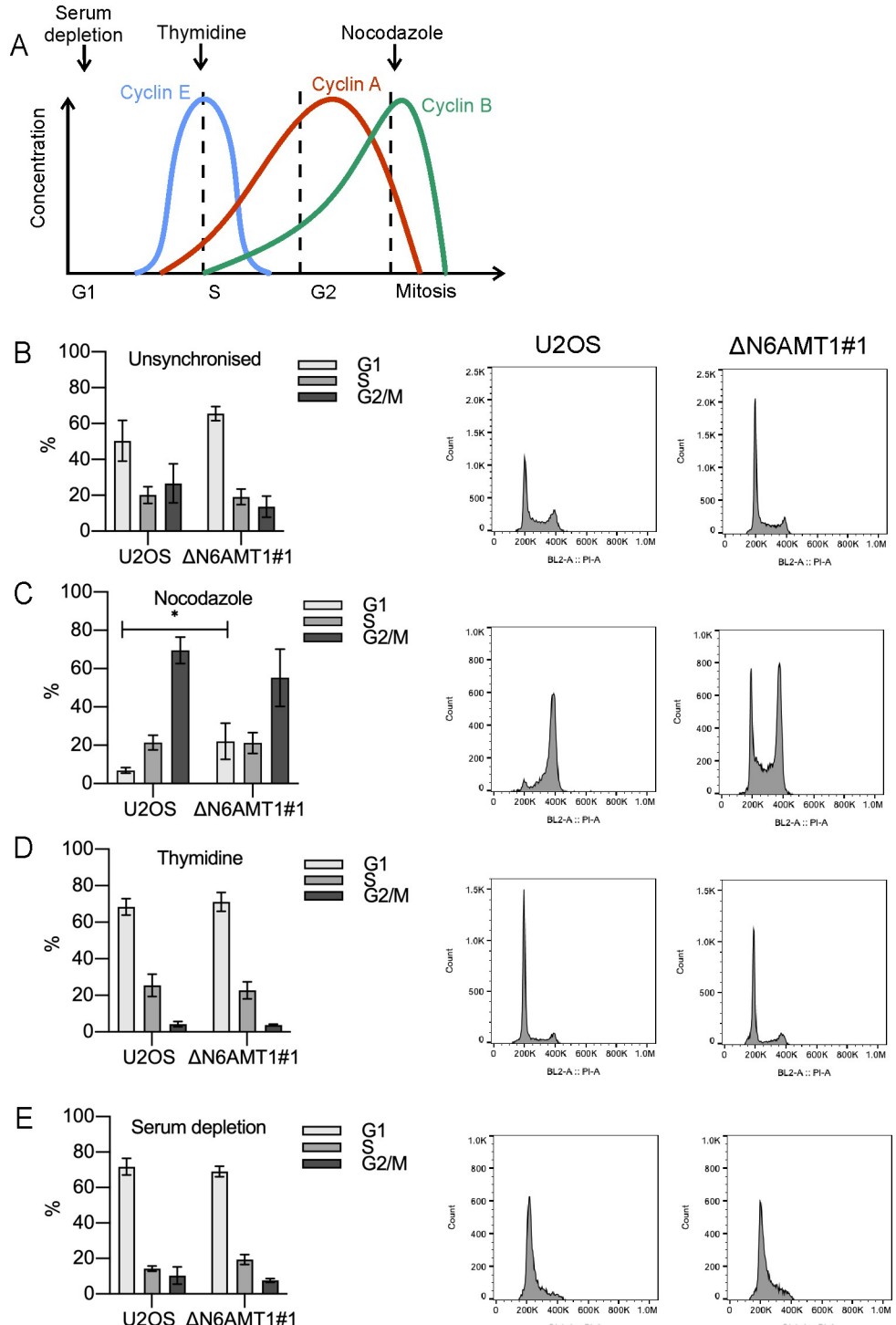

**Fig 2. Cell cycle distribution of U2OS and N6AMT1-depleted cells.** (A) Schematic representation of Cyclin A, B and E expression levels during the cell cycle. Serum depletion and thymidine and nocodazole treatment were used to arrest the cells at the indicated cell cycle points. (B-E) Cell cycle phase distribution of the U2OS and N6AMT1-knockout cell lines after the indicated treatments was determined by propidium iodide staining and flow cytometry analysis. The average of three biologically independent samples is shown. Statistical analysis was performed with GraphPad Prism 8.4.3. The data are presented as the means ± SDs; *P < 0,05, **P < 0,001, ***P < 0,0001 according to multiple t tests.

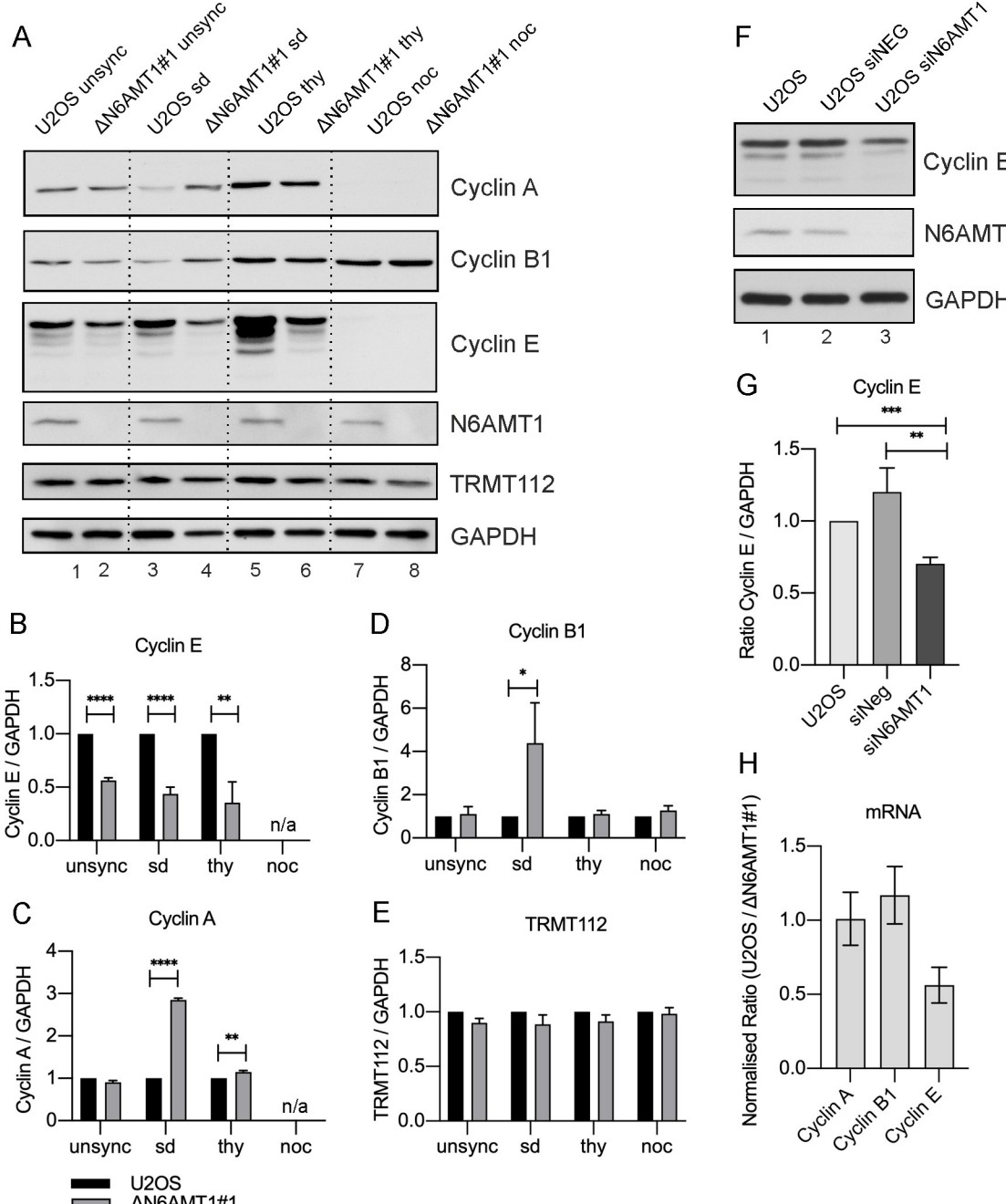

**Fig 3. N6AMT1 impacts the cyclin E levels.** (A) Unsynchronized cells and cells that were synchronized with serum depletion, thymidine treatment and nocodazole treatment were analysed via Western blotting with antibodies against Cyclin A, Cyclin B1, Cyclin E, N6AMT1, TRMT112 and GAPDH. 10 μg of total protein was loaded per well. (B-E) Cyclin A, cyclin B, cyclin E and TRMT112 levels in U2OS and ΔN6AMT1#1 cells. Western blotting images were quantified using ImageJ, and the average of three biologically independent samples is shown. Statistical analysis was performed with GraphPad Prism 8.4.3. The data are presented as the means ± SDs; *P < 0,05, **P < 0,001, ***P < 0,0001 according to multiple t tests. (F) Untreated cells and cells that were treated with the indicated siRNAs were analysed via Western blotting with antibodies against Cyclin E, N6AMT1 and GAPDH. (G) Cyclin E levels in U2OS cells after siRNA treatment. Western blotting images were quantified using ImageJ, and the average of three biologically independent samples is shown. Statistical analysis was performed with GraphPad Prism 8.4.3. The data are presented as the means ± SDs; *P < 0,05, **P < 0,001, ***P < 0,0001 according to multiple t tests. (H) qRT–PCR verification of cyclin A, cyclin B and cyclin E levels in ΔN6AMT1#1 cells. The average of three biologically independent samples is shown. Statistical analysis was performed with GraphPad Prism 8.4.3. The data are presented as the means ± SDs; *P < 0,05, **P < 0,001, ***P < 0,0001 according to multiple t tests.

methyltransferase activity, between U2OS and ΔN6AMT1#1 cells under any of the tested conditions (Fig 3A and 3E). The downregulation of cyclin E levels was also detected in partial HEK293 N6AMT1 k/o cells (S1C and S1F Fig). In addition, cyclin B levels were higher in serum depleted and unsynchronised cells (S1C and S1E Fig).

To confirm that the observed reduction in the cyclin E levels in the ΔN6AMT1#1 cell line is specific and not due to potential off-target effects of the CRISPR/Cas9 system, an experiment using siRNAs to target the N6AMT1 gene was conducted. Three different siRNAs targeting the 3' UTR, exon 3 and exon 5 of the N6AMT1 gene were pooled and used to knock down N6AMT1. The results showed a clear reduction in cyclin E levels similar to that observed in the ΔN6AMT1#1 cell line (Fig 3F and 3G).

Previous studies have suggested that N6AMT1 may be involved in the regulation of the cell cycle at the mRNA level. To investigate this, qRT–PCR was performed to measure the mRNA levels of cyclin A, cyclin B, and cyclin E in the U2OS and ΔN6AMT1#1 cell lines. The normalized qRT–PCR data showed that the cyclin E mRNA levels were downregulated in the ΔN6AMT1#1 cell line, while the mRNA levels of cyclin A and cyclin B were similar to those in the U2OS cell line (Fig 3H). These results suggest that the reduction in the cyclin E levels observed in the N6AMT1-knockout cells occurs due to the downregulation of the cyclin E mRNA levels, both in U2OS and HEK293 cells (Fig 3H and S1B Fig).

## N6AMT1-depleted cells are highly sensitive to serum depletion

The cyclin E levels were clearly influenced by the absence of N6AMT1 (Fig 3A and 3B), and no difference was observed in the cyclin A and B levels in unsynchronized cells or thymidine- or nocodazole-treated cells (Fig 3A, 3C and 3D). However, clear differences were observed in serum-starved cells, as the cyclin A and cyclin B levels remained elevated in the ΔN6AMT1#1 cell line compared to the U2OS cell line (Fig 3A, Lanes 3–4, 3C and 3D).

To estimate whether the ΔN6AMT1#1 cell line is capable of persisting under starvation conditions and entering G0 phase arrest, cells were starved for up to 120 h (Fig 4A). U2OS cells showed no major changes in cell density at any of the analysed timepoints. Meanwhile, the ΔN6AMT1#1 cell line was highly sensitive to starvation. After 48 h of starvation, the cell index of the ΔN6AMT1#1 cell line started to decrease drastically, reaching almost the background level after 120 h of starvation (Fig 4A). During the experiment, no cell death was observed, but clear changes in the cell morphology were noticed. The reduced cell index reflects changes in the cell morphology and cell adhesion to the surface rather than to cell death.

To analyse the ability of cells to exit G0/G1 arrest, U2OS and ΔN6AMT1# cells were grown in serum-free medium for 24 hours, after which the cells were released by replacing the starvation medium with complete growth medium (Fig 4B). The cells were then monitored, and changes in the cyclin A, cyclin B and cyclin E levels were observed over time (Fig 4C–4G).

The results showed that in U2OS cells, the cyclin A, B, and E levels were undetectable after 24 hours of serum starvation, indicating that the cells had entered G0/G1 arrest (Fig 4C, Lane 1). However, 3 hours after release from arrest, all the analysed cyclin levels started to increase and continued to steadily rise until 24 hours post release, indicating classical cell cycle recovery (Fig 4C, Lanes 2–9, Fig 4E–4G).

In contrast, in ΔN6AMT1#1 cells, even after 24 hours of serum depletion, the cyclin A and cyclin B levels remained elevated, suggesting that these cells did not enter G0 phase arrest and instead continued to progress through the cell cycle (Fig 4D, Lanes 1–9). The western blot results from Fig 4C and 4D are normalised to GAPDH and represented graphically for cyclin A (Fig 4E), Cyclin B (Fig 4F) and Cyclin E (Fig 4G). The graphical representation of the cyclin

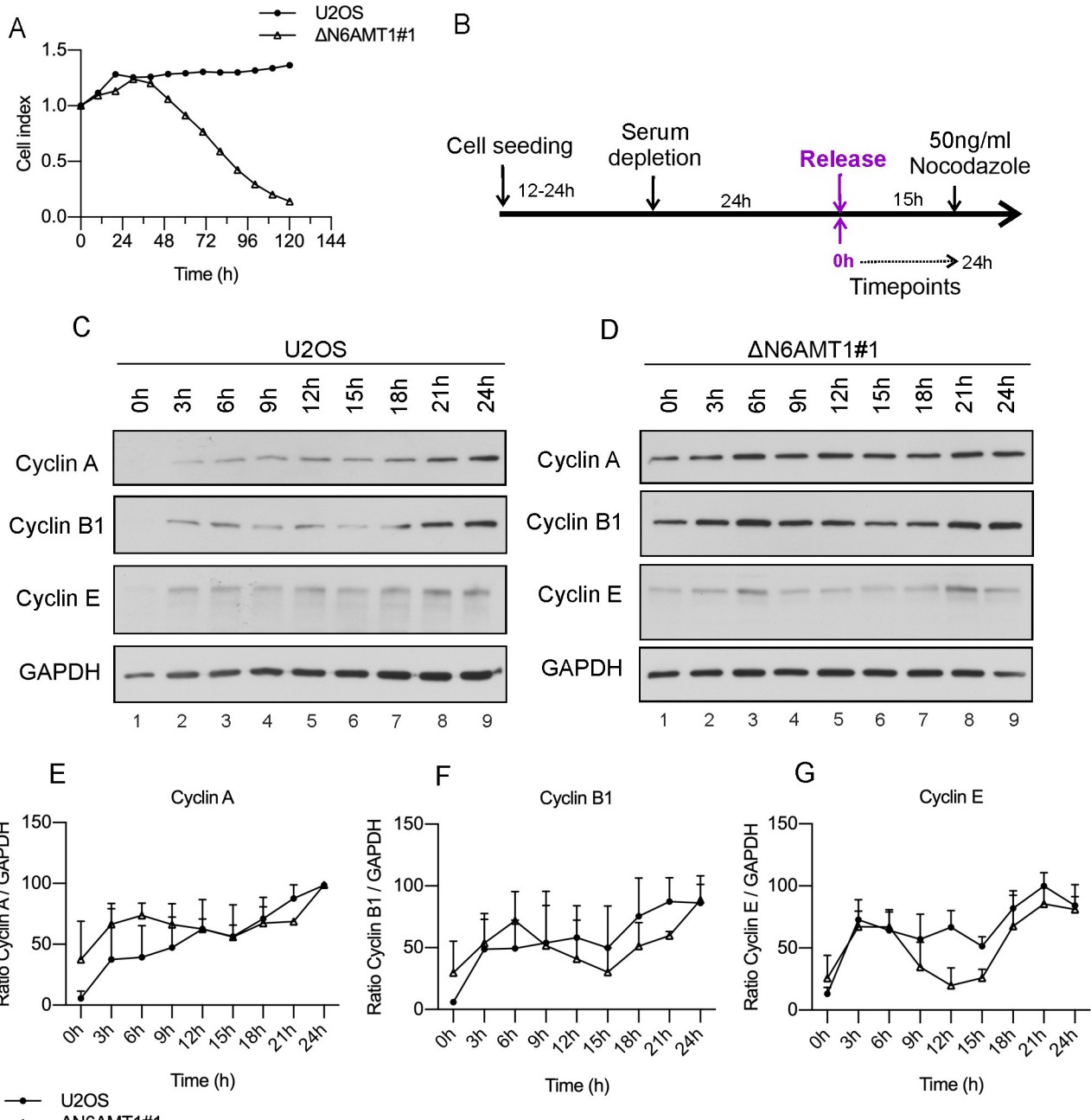

**Fig 4. Changes in the cyclin levels after serum depletion.** (A) The population density of the U2OS and ΔN6AMT1#1 cell lines in serum-depleted media was determined using an xCELLigence Real-Time cell analyser over a period of 120 h. (B) Schematic representation of the experimental setup for cell synchronization in the G1 phase. The cyclin A, B1 and E levels in (C) U2OS and (D) ΔN6AMT1#1 cells were analysed via Western blotting at the indicated timepoints. Cyclin A, B and E images were obtained after exposure time of 10 min for U2OS and 30 min ΔN6AMT1#1 samples. (E-G) Graphical representation of changes in the cyclin levels. Western blotting images were quantified using ImageJ, and the average of three biologically independent samples is shown.

levels illustrates that U2OS cells exhibited an expected recovery pattern, while ΔN6AMT1#1 cells did not exhibit G0 phase arrest and instead exhibited sustained high cyclin A and B expression levels throughout the experiment (Fig 4E–4G). Additionally, two peaks were observed in the cyclin B and cyclin E levels that were more noticeable in N6AMT1-depleted

cells (Fig 4C,4D, 4F and 4G), indicating that some cells remained in G0/G1 arrest longer than others.

## N6AMT1-depleted cells show delayed progression out of S phase arrest

U2OS and ΔN6AMT1# cells were synchronized in the S phase, first by seeding only mitotic cells followed by thymidine treatment; then, the cells were released to observe the progression of the cell cycle (Fig 5A). As expected, in the U2OS cells, the cyclin E levels remained high at 0–6 hours post release, and a subsequent decrease indicated progression into the G2 phase

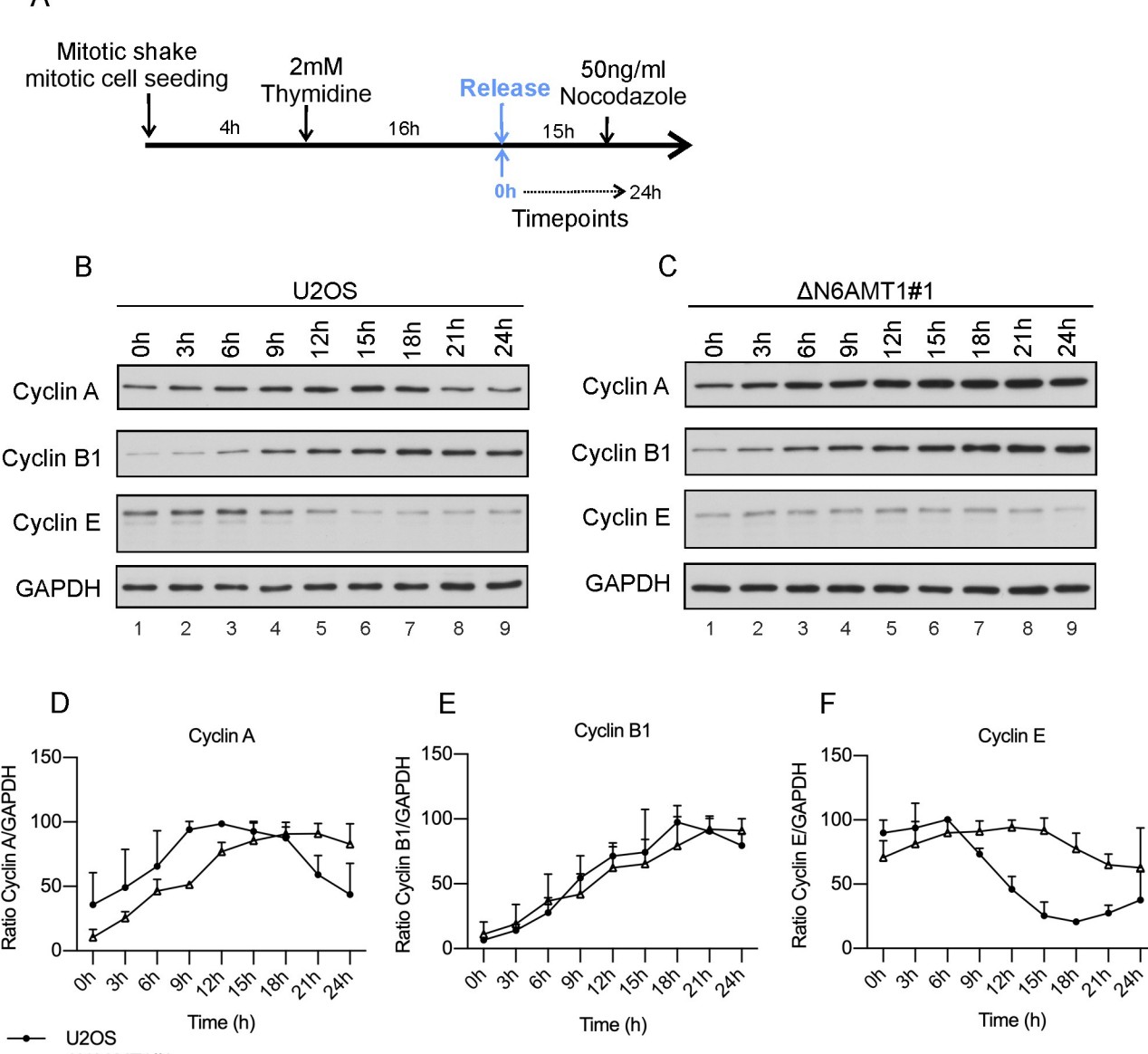

**Fig 5. Changes in cyclin levels after thymidine block.** (A) Schematic representation of the experimental setup for cell synchronization early in the S phase. The cyclin A, B1 and E levels in (B) U2OS and (C) ΔN6AMT1#1 were analysed via Western blotting at the indicated timepoints. Cyclin A, B and E images were obtained after exposure time of 2 min for U2OS and 15 min ΔN6AMT1#1 samples. (D-F) Graphical representation of changes in the cyclin levels. Western blotting images were quantified using ImageJ, and the average of three biologically independent samples is shown.

(Fig 5B and 5F). In contrast, ΔN6AMT1#1 cells did not exhibit decreases in the cyclin E levels, indicating a disruption in the normal progression of the cell cycle (Fig 5C and 5F).

The analysis of the cyclin A and cyclin B levels showed that U2OS cells progressed through the S phase in a typical manner, with the cyclin A levels increasing and reaching a plateau at 9 hours post release and then decreasing (Fig 5B and 5D). The cyclin B levels also increased and peaked at 18 hours post release (Fig 5B and 5E). In contrast, the increase in the cyclin A levels in ΔN6AMT1#1 cells was delayed by 9 hours compared to that in U2OS cells, while the increase in the cyclin B levels was not significantly delayed (Fig 5C–5E).

These results suggest that N6AMT1 depletion in cells causes a defect in the normal progression through the S phase during the cell cycle, potentially due to a disruption in the regulation of cyclin E levels.

## N6AMT1 depletion affects the normal progression of mitosis

U2OS and ΔN6AMT1# cells were first treated with nocodazole, and after the cells reached prophase, they were subjected to a mitotic shake and seeded in complete growth media to release them from the block. Samples were collected over time to track changes in the expression levels of different cyclins (Fig 6A). The results showed that U2OS cells exhibited an expected pattern of cell cycle progression after release from the nocodazole block, with cyclin A and cyclin B expression levels sharply decreasing after the 0 h time point and then increasing again 12 h later (Fig 6B, 6D and 6E). Cyclin E began to accumulate 6 hours after release, which is consistent with its known role in promoting the G1/S transition (Fig 6B, Lanes 5–9, Fig 6F). In ΔN6AMT1#1 cells, the expression levels of cyclin B1 slowly decreased over 2 hours after release and were detectable at low levels throughout the experiment (Fig 6C and 6E). In contrast to U2OS cells, the levels of cyclin A were much higher at the 0-hour time point and remained detectable 2 hours after release (Fig 6C and 6D). No difference was observed in the trend of cyclin E expression in N6AMT1-depleted cells (Fig 6B, 6C and 6E).

In summary, the depletion of N6AMT1 in U2OS cells reduced the cell proliferation rate and disrupted the cell cycle by causing a delay in the progression of mitosis and the S phase. Additionally, these cells were highly sensitive to serum starvation. In ΔN6AMT1#1 cells, the protein expression of cyclin E was downregulated throughout the cell cycle, and it was confirmed that this downregulation is regulated at the transcriptional level. Additionally, there were no differences in cyclin A or cyclin B expression, with the exception of the starved cell population, in which the cyclin A and B levels stayed elevated.

## Discussion

N6AMT1 is a well-described protein methyltransferase that is functional only as a heterodimer with TRMT112 [8, 9, 29]. Multiple studies have associated N6AMT1 with different conditions, such as tumour progression, diabetes, neuronal disorders and mouse embryonic lethality [11, 14, 16, 17, 24]. The precise mechanism of how N6AMT1 impacts these phenotypes is unknown.

N6AMT1 has been shown to be important for normal cell growth and proliferation in prokaryotes, yeast and immortalized cell lines. In accordance with previously published data, the current study confirms that the cell proliferation rate is significantly reduced in N6AMT1-depleted cells and that the accumulation of these cells in the G1 phase indicates that N6AMT1 might be involved in cell cycle regulation. Interestingly, the RNA-Seq or ChIP-Seq data from two research groups show that multiple cell cycle-related genes are either up- or downregulated in N6AMT1-depleted cells [7, 11, 24]. These studies highlighted different genes that are associated with cell cycle regulation, including RB1,

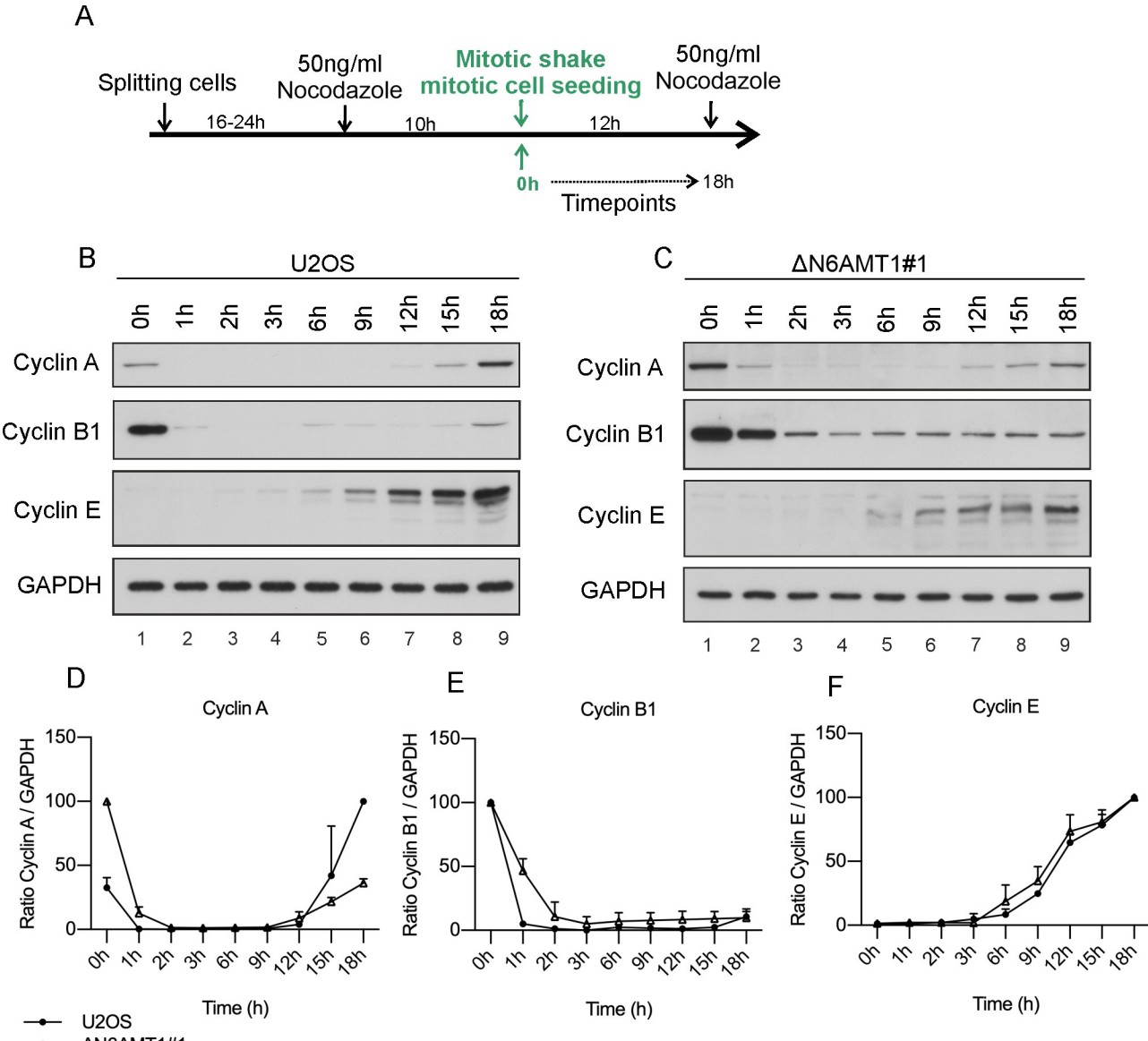

**Fig 6. Changes in cyclin levels after nocodazole arrest.** (A) Schematic representation of the experimental setup for cell synchronization early in prophase. (B) The cyclin A, B1 and E levels in U2OS and (C) ΔN6AMT1#1 cells were analysed via Western blotting at the indicated timepoints. Cyclin A, B and E images were obtained after 1h exposure time for both cell lines. (D-F) Graphical representation of changes in the cyclin levels. Western blotting images were quantified using ImageJ, and the average of three biologically independent samples is shown.

CDK6, CNBP, and p53. The current study further confirms that N6AMT1 is indeed involved in cell cycle regulation in multiple ways.

In U2OS cells, the depletion of N6AMT1 led to a significant reduction in the cyclin E levels. Interestingly, the effect was not specific to certain cell cycle phases, as the cyclin E level was consistently lower in all cell cycle phases that were analysed. This suggests that the regulation of the cyclin E levels affects the entire cell life cycle rather than being linked to specific cell cycle phases or individual dysfunctional cell cycle checkpoints. Moreover, the classic pattern of up- and downregulation of cyclin E [46] was disrupted in N6AMT1-depleted cells. When released from thymidine-induced arrest, the levels of cyclin E failed to decrease as they did in

U2OS cells, indicating a prolonged G1 phase. It could be speculated that the depletion of N6AMT1 might have caused the cyclin E levels reach a minimum threshold that is required for sustaining cell growth/cycle, leaving no margin for further reduction.

Although it was confirmed that cyclin E downregulation occurs at the mRNA level, the exact mechanism by which N6AMT1 is involved in cyclin E transcriptional regulation requires further investigation. N6AMT1 is a known methyltransferase that has previously been associated with H4K12 methylation and the cell proliferation rate [7]. It is possible that N6AMT1-mediated histone 4 methylation also regulates the transcription of cyclin E. Furthermore, N6AMT1 has been shown to methylate a variety of proteins that have not yet been fully described. This suggests that N6AMT1 could regulate cyclin E transcription through the methylation of certain transcription factors. Another possibility to consider is the role of N6AMT1 in DNA methylation. Depletion of N6AMT1 may lead to changes in the DNA methylation pattern of the cyclin E gene region, potentially influencing its transcription.

The current study also revealed a delay in the progression of mitosis following release from nocodazole-induced arrest in N6AMT1-depleted cells. The protein levels of cyclin A and cyclin B1 decreased almost 2 hours later in N6AMT1-depleted cells than in U2OS cells, indicating a potential regulatory function of N6AMT1 in mitosis. It is possible that the observed effect in ΔN6AMT1#1 cells might still be a consequence of the reduced levels of cyclin E, as no differences in the total protein levels of cyclin A and cyclin B were observed in the mitotic cell population. However, N6AMT1 could perform an additional function in mitosis that is unrelated to the regulation of cyclin E.

In the current study, N6AMT1-depleted cells were highly sensitive to serum depletion. The sensitivity was demonstrated by changes in the cell morphology and reduced proliferation and adhesion of the cells on the surface rather than cell death. No effect on U2OS cell morphology and adhesion was observed upon serum depletion. A previous study demonstrated that cyclin E-deficient cells actively proliferate but are unable to re-enter the cell cycle from the G0 phase [47]. This finding implies that in N6AMT1-depleted cells, due to the reduced cyclin E levels, enters the G0 phase more efficiently than U2OS cells, indicating that N6AMT1 may impact cell entry/exit from the resting phase. This may indicate that N6AMT1 plays a role in the ability of cancer cells to enter quiescence, which is a mechanism that allows cancer cells to become resistant to chemotherapy and is associated with cancer recurrence. Therefore, the level of N6AMT1 expressed in patients may have the potential to be used as a biomarker for measuring cancer recurrence or treatment success rates. However, the current study focused only on serum-depleted starvation; therefore, the possibility that the phenomena observed here may be specific to the used method cannot be excluded. Further investigation is required to confirm this phenomenon by using different quiescence induction methods and distinguishing quiescent cells with specific markers from normal G1 phase cells.

The current study, along with previous studies, indicates that N6AMT1 is a multifunctional protein that plays a role in regulating cell growth, cancer progression and embryonic development. To our knowledge, this is the first time that N6AMT1 has been shown to be involved in cell cycle regulation from multiple perspectives, such as regulating cyclin E levels, controlling mitotic progression, and modulating quiescence entry. Future research should focus on determining whether these cell cycle disruptions are due to different N6AMT1 functions or simply a consequence of cyclin E downregulation. N6AMT1 may regulate the cell cycle directly, indirectly, or as part of larger or smaller, not-yet-described complexes. Understanding the exact mechanisms of how N6AMT1 functions at so many different levels will be a topic of future and ongoing research.

## Supporting information

**S1 Fig. N6AMT is important for cell proliferation and cyclin E regulation in HEK293 cells.**
(A) The proliferation rate of the U2OS, ΔN6AMT1#1 and ΔN6AMT1#2 cell lines were determined using an xCELLigence Real-Time cell analyser, and data were recorded for a total of 7 days. (B) qRT–PCR verification of cyclin A, cyclin B and cyclin E levels in HEK293ΔN6AMT1 cells. The average of two biologically independent samples is shown. Statistical analysis was performed with GraphPad Prism 8.4.3. (C) Unsynchronized cells and cells that were synchronized with serum depletion and thymidine treatment were analysed via Western blotting with antibodies against Cyclin A, Cyclin B1, Cyclin E, N6AMT1 and GAPDH. 10 μg of total protein was loaded per well. (D-F) Cyclin A, cyclin B and cyclin E levels in U2OS and ΔN6AMT1#1 cells. Western blotting images were quantified using ImageJ, and the average of three biologically independent samples is shown. Statistical analysis was performed with GraphPad Prism 8.4.3. The data are presented as the means ± SDs; *P < 0,05, **P < 0,001, ***P < 0,0001 according to multiple t tests.
(PDF)

## Author Contributions

**Conceptualization:** Margit Mutso, Reet Kurg.

**Data curation:** Margit Mutso.

**Formal analysis:** Margit Mutso, Baiba Brūmele.

**Funding acquisition:** Reet Kurg.

**Investigation:** Baiba Brūmele, Evgeniia Serova, Fred Väärtnõu, Mihkel Suija.

**Methodology:** Margit Mutso.

**Supervision:** Margit Mutso, Reet Kurg.

**Visualization:** Baiba Brūmele.

**Writing – original draft:** Margit Mutso.

**Writing – review & editing:** Margit Mutso, Baiba Brūmele, Reet Kurg.

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
