## [Decision Letter · Decision Letter 0]

13 Dec 2023

PONE-D-23-27992The methyltransferase N6AMT1 participates in the cell cycle by regulating cyclin E levelsPLOS ONE

Dear Dr. Mutso,

Thank you for submitting your manuscript to PLOS ONE. After careful consideration, we feel that it has merit but does not fully meet PLOS ONE’s publication criteria as it currently stands. Therefore, we invite you to submit a revised version of the manuscript that addresses the points raised during the review process.

We look forward to receiving your revised manuscript.

Kind regards,

Yi-Ren Hong

Academic Editor

PLOS ONE

Journal Requirements:

The name of the colleague or the details of the professional service that edited your manuscript.A copy of your manuscript showing your changes by either highlighting them or using track changes (uploaded as a *supporting information* file).A clean copy of the edited manuscript (uploaded as the new *manuscript* file).

"This research was funded by the Institute of Technology basic funding grant PLTTI20915, award to RK and Estonian Research Council grant PRG1506, award to RK."

**Additional Editor Comments:**

This manuscript discovered that in N6AMT1-depleted cells, the transcription of cyclin E was downregulated, which indicates that N6AMT1 is involved in the regulation of cyclin E transcription. The approached method and data appears interesting. However, some concerns are raised:

1. In Fig2A, you normalize the cyclin A, B, E by their concentration, but all the concentration level of Cyclin A, B, & E appeared the same. What are the data you rely on to calculate them on this manuscript?

2. Fig.4 C &D (Knockdown N6AMT1) Cyclin E showed not very significantly dismiss. Also, same as Fig. 5B & C (Knockdown N6AMT1). On the contract, the data showed that Cyclin A & B seem increase and prolong longer, need to explain more carefully. In this regarding, how you draw your conclusion to cyclin E was downregulated on G0/G1 and S which cause cell cycle disrupted or arrested.

3. What will be expected for cycle D1 (G1 phase marker)?

4. This manuscript deals with epigenetic tools which contain: The Writers, The Readers and The Erasers. Thus, better to mention the general roles of methyltransferase N6AMT1 (such as in addition to Cyclin E, any other reader for N6AMT1; Does N6AMT1 have mutant form as dominant negative for control?) in discuss section.

Reviewers' comments:

Reviewer's Responses to Questions

**Comments to the Author**

1. Is the manuscript technically sound, and do the data support the conclusions?

Reviewer #1: Yes

2. Has the statistical analysis been performed appropriately and rigorously? 

Reviewer #1: Yes

3. Have the authors made all data underlying the findings in their manuscript fully available?

Reviewer #1: Yes

4. Is the manuscript presented in an intelligible fashion and written in standard English?

Reviewer #1: Yes

5. Review Comments to the Author

Reviewer #1: The article “The methyltransferase N6AMT1 participates in the cell cycle by regulating cyclin E levels”, authors aim to evaluate the biological functions of N6AMT1, and they demonstrate that N6AMT1 is involving in cell cycle regulation through modulating cyclin E expression. Because N6AMT1 has been reported to participate in tumor progression; however, the detail mechanisms are still unclear. This is the first study to demonstrate that N6AMT1 participates in the expression of cyclin E and modulating cell cycle. In addition, the evidences provide in this study is convincing. Therefore, it is recommended to publish this interesting manuscript in the PLOS ONE.

Comments:

1. It is interesting to investigate whether this finding is common in various cell types. If authors can provide more types of cells to verify this finding is more expectant.

2. In the clinical, cancer patients treated with anti-cancer agents, and the tumor quiescence may cause recurrence. Therefore, the expressions of N6AMT1 and tumor quiescence as well as tumor recurrence or drug resistance is the questions worth discussing.

6. PLOS authors have the option to publish the peer review history of their article (what does this mean?). If published, this will include your full peer review and any attached files.

Reviewer #1: No

---

## [Author Response · Author response to Decision Letter 0]

19 Jan 2024

We are very thankful to the Academical Editor and Reviewer for the comments. We have included answers to current document and revised our manuscript.

Concerns raised by the Academic Editor:

1. In Fig2A, you normalize the cyclin A, B, E by their concentration, but all the concentration level of Cyclin A, B, & E appeared the same. What are the data you rely on to calculate them on this manuscript?

Answer: Fig 2A is purely illustrative figure to precent schematically well-established knowledge; the fluctuation of different cyclin levels in the cell cycle and indicate where treatments, used in the current study, block in the context of cell cycle.

In the study we analyse the cell cycle in U2OS cells and investigate how the depletion of N6AMT1 affects the cell cycle. The calculation of cyclin levels is normalised to GAPDH and we only compare the levels of cyclin-GAPDH ratio in U2OS cells and N6AMT1 knock out cells. 

2. Fig.4 C &D (Knockdown N6AMT1) Cyclin E showed not very significantly dismiss. Also, same as Fig. 5B & C (Knockdown N6AMT1). On the contract, the data showed that Cyclin A & B seem increase and prolong longer, need to explain more carefully. In this regarding, how you draw your conclusion to cyclin E was downregulated on G0/G1 and S which cause cell cycle disrupted or arrested.

Answer: The experiments presented on the Fig. 4, 5 and 6 are shown to analyse the fluctuation of cyclin levels after cells are released from certain arrest. The samples from U2OS and N6AMT1 k/o are run on different gels and the exposure times of the western blots vary considerably. The condition that suited the best for U2OS, resulted too weak bands for N6AMT1 k/o cells. Therefore, we cannot compare the band of Fig 4C and D or Fig. 5B and C. Normalisation to GAPDH is used to compare different expression levels and the behaviour of cyclin levels between U2OS and N6AMT1 k/o cells was compared. 

In Fig. 4 we compare how quickly does U2OS and N6AMT1 k/o cells recover from starvation. In Fig 4C, the exposure times of U2OS western blot images are 10 min for cyclin A, B and E but in Fig. 4D the N6AMT1 k/o western blot images exposure time is 30 min for the analysed cyclins. The results show that in N6AMT1 k/o cells cyclin A and B levels are higher than in U2OS cells (Fig 4C, lane1 and D lane 1) which is repeating the Fig.3 results (Fig.3A, C, D). We also see that in case of U2OS all cyclin levels are raising as they progress out of G1 arrest. Meanwhile in N6AMT1 k/o cells we see only very mild fluctuation of cyclin A and B. This refers that N6AMT1 k/o cells progression out of G0/G1 phase is disturbed. 

In Fig 5. we compare how in U2OS and N6AMT1 k/o cells the cyclin levels change after release from thymidine block (corresponds to S phase block). For U2OS western blot the exposure time is 2 min and for N6AMT1 k/o cells western blot exposure time is 15 min. The cyclin E level changes in U2OS cells is presented on the western blot data Fig. 5B and additionally on Fig. 5F (the round dot line). For U2OS cells the cyclin E levels drop as the cells are progressing out of S phase. Meanwhile for the N6AMT1 k/o cells there is no fluctuation on cyclin E levels Fig. 5C and F (the line with open triangles). 

In Fig 6. the exposure time of all the cyclins in U2OS and N6AMT1 k/o cells is 1h. 

In case of cyclin E, we do not see differences between U2OS and N6AMT1 k/o cells when they progress out of mitosis (Fig. 6E), but we see differences when cells are released from the thymidine (Fig. 5F) or serum depletion block (Fig. 4G). This refers that the cell cycle progress is interrupted in G0/G1 and/or S phase. In Fig. 3A and B is demonstrated that in N6AMT1 k/o cells, cyclin E levels are downregulated in all analysed cell phases, except mitosis where it is under detection limit for both U2OS and N6AMT1 k/o cells.

To clarify this, we have added exposure times to figures legends (Fig 4, 5 and 6) and sentence to manuscript in lane 354-356 and in.

Lane 354-356

“The western blot results from Fig 4C and D are normalized to GAPDH and represented graphically for cyclin A (Fig 4E), Cyclin B (Fig 4F) and Cyclin E (Fig 4G).”

Fig 4. Changes in the cyclin levels after serum depletion. 

(A) The population density of the U2OS and ΔN6AMT1#1 cell lines in serum-depleted media was determined using an xCELLigence Real-Time cell analyser over a period of 120 h. (B) Schematic representation of the experimental setup for cell synchronization in the G1 phase. The cyclin A, B1 and E levels in (C) U2OS and (D) ΔN6AMT1#1 cells were analysed via Western blotting at the indicated timepoints. Cyclin A, B and E images were obtained after exposure time of 10 min for U2OS and 30 min ΔN6AMT1#1 samples. (E-G) Graphical representation of changes in the cyclin levels. Western blotting images were quantified using ImageJ, and the average of three biologically independent samples is shown.

Fig 5. Changes in cyclin levels after thymidine block. 

(A) Schematic representation of the experimental setup for cell synchronization early in the S phase. (B) The cyclin A, B1 and E levels in (B) U2OS and (C) ΔN6AMT1#1 were analysed via Western blotting at the indicated timepoints. Cyclin A, B and E images were obtained after exposure time of 2 min for U2OS and 15 min ΔN6AMT1#1 samples. (D-F) Graphical representation of changes in the cyclin levels. Western blotting images were quantified using ImageJ, and the average of three biologically independent samples is shown.

Fig 6. Changes in cyclin levels after nocodazole arrest. 

(A) Schematic representation of the experimental setup for cell synchronization early in prophase. (B) The cyclin A, B1 and E levels in U2OS and (C) ΔN6AMT1#1 cells were analysed via Western blotting at the indicated timepoints. Cyclin A, B and E images were obtained after 1h exposure time for both cell lines. (D-F) Graphical representation of changes in the cyclin levels. Western blotting images were quantified using ImageJ, and the average of three biologically independent samples is shown.

3. What will be expected for cycle D1 (G1 phase marker)?

Answer: 

In the current study we have included only the cyclin A, cyclin B and cyclin E as they are classical cyclins used to distinguish the different cell cycle phases. We have not tested the levels of cyclin D1 but as all cyclin levels are at some conditions affected, we expect to find some differences as well for cyclin D1 levels. Cylin D1 and other cell cycle relevant proteins are analysed in future studies.

4. This manuscript deals with epigenetic tools which contain: The Writers, The Readers and The Erasers. Thus, better to mention the general roles of methyltransferase N6AMT1 (such as in addition to Cyclin E, any other reader for N6AMT1; Does N6AMT1 have mutant form as dominant negative for control?) in discuss section.

Answer: Current knowledge about N6AMT1 is too limited to make any final conclusions or statements about N6AMT1 as epigenetic writer. Classical epigenetic tools are connected to histone modification or DNA modification. Indeed, it has confirmed that N6AMT1-TMRT112 complex is H4K12 methyltransferase. This knowledge is included in the introduction (lane 31-37) and discussion (lane 449-454) section. N6AMT1 DNA methylation function has so far very controversial studies as we have discussed it in lane 38-49 and 456-458. It has been described by different studies that N6AMT1 is important for cell cycle and cell proliferation, and from the RNAseq data from previous studies have suggested effect on RB1, CDK6, CNBP, and p53 among other cell cycle related proteins, which all might be the readers of N6AMT1 but this needs to have further studies to confirm as discussed in lane 432-437. 

There is no information about the dominant negative form of N6AMT1 isoform 1 for the full-length functional protein. The N6AMT1 isoform 2 that lacks the binding motif for its partner TMRT112 does not function as methyltransferase. Also, it has been demonstrated to be unstable and the levels of endogenous N6AMT1 isoform 2 is undetectable in the cells. N6AMT1 is involved in many different cellular and molecular functions: a protein methyltransferase (methylating both lysine and glutamine), DNA methyltransferase and probably it has some other activities not described yet. Using the dominant negative form for one N6AMT1 activity may not be the same for its other functions.

Reviewer comments.

1. It is interesting to investigate whether this finding is common in various cell types. If authors can provide more types of cells to verify this finding is more expectant.

Answer: We have analysed the role of N6AMT1 in noncancerous HEK293 cells. We have included Supplementary figure 1, demonstrating that additionally to U2OS cells, the same effect is seen in HEK293 cell line. In case of HEK293 cell, we managed to obtain only a partial N6AMT1 knock out cell line. Regardless analysing numerous colonies only partial N6AMT1 knock out cells were viable. HEK293 cells are non-cancerous and therefore N6AMT1 may play more vital role compared to U2OS cells, which are able to compensate the absence of N6AMT1. Nevertheless, we can detect the same pattern in partial HEK293 knock-out cells similar to U2OS knock-out cells. Firstly, the cells growth rate was severely reduced in HEK293 ΔN6AMT1 cells (Fig. S1E). We detected the reduced level of cyclin E protein in HEK293 ΔN6AMT1 in serum depleted and thymidine treated sample in normalized western blot assay (Fig S1A, E). Similarily to U2OS cells, in the HEK293 ΔN6AMT1cells the cyclin B1 levels were increased in serum depleted samples (Fig S1A, C). The cyclin E was affected on mRNA levels (Fig S1B) which repeats the results obtained in U2OS cells.

Corresponding row have been included to manuscript results section: 

Lane 206-207 “ The cell growth rate reduction was also detected in partial N6AMT1 k/o HEK293 cells (Fig S1A).”

Lane 276-278

„ The downregulation of cyclin E levels was also detected in partial HEK293 N6AMT1 k/o cells (Fig S1C, F). In addition, cyclin B levels were higher in serum depleted and unsynchronised cells (Fig S1C, E).”

Lanes 314 

„These results suggest that the reduction in the cyclin E levels observed in the N6AMT1-knockout cells occurs due to the downregulation of the cyclin E mRNA levels, both in U2OS and HEK293 cells (Figs 3H and S 1B).“

Lanes 637-650 Supplementary figure Legend is added.

Supporting information

S1 Fig. N6AMT is important for cell proliferation and cyclin E regulation in HEK293 cells

(A) The proliferation rate of the U2OS, ΔN6AMT1#1 and ΔN6AMT1#2 cell lines were determined using an xCELLigence Real-Time cell analyser, and data were recorded for a total of 7 days. (B) qRT‒PCR verification of cyclin A, cyclin B and cyclin E levels in HEK293ΔN6AMT1 cells. The average of two biologically independent samples is shown. Statistical analysis was performed with GraphPad Prism 8.4.3. (C) Unsynchronized cells and cells that were synchronized with serum depletion and thymidine treatment were analysed via Western blotting with antibodies against Cyclin A, Cyclin B1, Cyclin E, N6AMT1 and GAPDH. 10 µg of total protein was loaded per well. (D-F) Cyclin A, cyclin B and cyclin E levels in U2OS and ΔN6AMT1#1 cells. Western blotting images were quantified using ImageJ, and the average of three biologically independent samples is shown. Statistical analysis was performed with GraphPad Prism 8.4.3. The data are presented as the means ± SDs; *P < 0,05, **P < 0,001, ***P < 0,0001 according to multiple t tests.

2. In the clinical, cancer patients treated with anti-cancer agents, and the tumour quiescence may cause recurrence. Therefore, the expressions of N6AMT1 and tumour quiescence as well as tumour recurrence or drug resistance is the questions worth discussing.

Answer: This point is discussed in lanes 466-474. The text is modified to emphasis the clinical aspect.

Lane 477-479 „Therefore, the level of N6AMT1 expressed in patients may have the potential to be used as a biomarker for measuring cancer recurrence or treatment success rates.“

---

## [Editor Report · Decision Letter 1]

1 Feb 2024

The methyltransferase N6AMT1 participates in the cell cycle by regulating cyclin E levels

PONE-D-23-27992R1

Dear Dr. Mutso,

We’re pleased to inform you that your manuscript has been judged scientifically suitable for publication and will be formally accepted for publication once it meets all outstanding technical requirements.

Kind regards,

Yi-Ren Hong

Academic Editor

PLOS ONE

Additional Editor Comments (optional):

I go over this revision, and I am satisfied with the explanations and additional informations provided by the authors. I can recommend this manuscript for acceptance by the Plos One.
---

## [Editor Report · Acceptance letter]

13 Feb 2024

PONE-D-23-27992R1 

PLOS ONE

Dear Dr. Mutso, 

I'm pleased to inform you that your manuscript has been deemed suitable for publication in PLOS ONE. Congratulations! Your manuscript is now being handed over to our production team.

Kind regards, 

on behalf of

Professor Yi-Ren Hong 

Academic Editor

PLOS ONE